# The Potential of Fasting-Mimicking Diet as a Preventive and Curative Strategy for Alzheimer’s Disease

**DOI:** 10.3390/biom13071133

**Published:** 2023-07-14

**Authors:** Virginia Boccardi, Martina Pigliautile, Anna Giulia Guazzarini, Patrizia Mecocci

**Affiliations:** 1Department of Medicine and Surgery, Institute of Gerontology and Geriatrics, University of Perugia, Piazzale Gambuli 1, 06132 Perugia, Italy; martina.pigliautile@gmail.com (M.P.); annagiuliaguazzarini@gmail.com (A.G.G.); patrizia.mecocci@unipg.it (P.M.); 2Division of Clinical Geriatrics, NVS Department, Karolinska Institutet, 17177 Stockholm, Sweden

**Keywords:** Alzheimer, diet, health span, nutrition, prevention

## Abstract

This review examines the potential of fasting-mimicking diets (FMDs) in preventing and treating Alzheimer’s disease (AD). FMDs are low-calorie diets that mimic the physiological and metabolic effects of fasting, including the activation of cellular stress response pathways and autophagy. Recent studies have shown that FMDs can reduce amyloid-beta accumulation, tau phosphorylation, and inflammation, as well as improve cognitive function in animal models of AD. Human studies have also reported improvements in AD biomarkers, cognitive functions, and subjective well-being measures following FMDs. However, the optimal duration and frequency of FMDs and their long-term safety and efficacy remain to be determined. Despite these uncertainties, FMDs hold promise as a non-pharmacological approach to AD prevention and treatment, and further research in this area is warranted.

## 1. Introduction

With the growing older population, there is a notable rise in the prevalence of dementia worldwide. Alzheimer’s disease (AD) continues to be the leading cause of dementia, impacting approximately 50 million individuals globally, primarily among older people [1]. AD is a heterogeneous and multifactorial disorder characterized by a progressive decline in cognitive functions, behavioral problems, and poor self-care, leading to progressive disability [2,3]. In detail, AD is a progressive neurodegenerative disorder that manifests along a continuum, from subjective cognitive decline (SCD) to mild cognitive impairment (MCI), eventually leading to dementia. SCD refers to self-reported cognitive complaints in individuals who are otherwise functioning normally and do not show any objective cognitive impairment. These individuals may notice subtle changes in their memory or cognitive abilities but can still perform daily activities without significant difficulties. MCI represents a stage where individuals experience noticeable cognitive decline beyond what is expected for their age and education level. It is characterized by mild impairments in memory, thinking, and/or other cognitive domains. However, individuals with MCI can still carry out their daily activities independently without substantial functional impairment. MCI can be further classified into two subtypes: amnestic MCI (primarily affecting memory) and non-amnestic MCI (affecting domains other than memory). Dementia is the most advanced stage of the AD continuum, where individuals experience significant cognitive decline that impairs their ability to function independently. Memory loss, confusion, language difficulties, and challenges in problem-solving and decision-making are commonly observed in dementia. As the disease progresses, individuals may require assistance with daily activities, and their cognitive and functional impairments become more severe [2].

From a pathological point of view, AD is characterized by the progressive deposition of amyloid β-peptide (Aβ) as amyloid plaques, hyperphosphorylated tau protein intracellularly as neurofibrillary tangles (NFTs), and neuronal loss [3]. AD can be divided into two main categories. The first is early onset AD (EOAD), which typically affects younger individuals mainly due to mutations in specific genes responsible for amyloid protein processing. The second is late-onset AD (LOAD), diagnosed in older individuals and mainly associated with lifestyle, environmental, and disease risk factors [4].

Despite over one hundred years of research, currently, there is no effective treatment to prevent AD development. However, a growing and recent body of evidence has identified potentially modifiable risk factors for AD and related dementias, including hypertension, obesity, diabetes, depression, cigarette smoking, hearing loss, and binge drinking [5,6]. Therefore, interventions targeting each risk factor and mechanism in the meantime are required for an optimal preventive effect. The Finnish Geriatric Intervention Study to Prevent Cognitive Impairment and Disability (FINGER) is the first large, long-term trial showing that a multidomain lifestyle-based intervention ameliorates vascular and lifestyle-related risk factors as well as preserves cognitive functioning reducing the incidence of cognitive decline among older persons [7,8]. Based on current understanding, it is thought that diet can impact the risk of developing AD by modulating metabolic and energetic pathways. For example, excessive body weight resulting from a diet high in carbohydrates and fats has been linked to an increased risk of AD. Conversely, reducing calorie intake in a regular diet can extend a healthy lifespan and decrease the incidence of AD and related dementia [9]. For this purpose, several dietary patterns have been proposed, such as the Mediterranean diet, the DASH (dietary approaches to stop hypertension) diet, and the MIND diet (Mediterranean-DASH intervention for neurodegenerative delay diet) [10]. These studies collectively show that minimizing the intake of trans fatty acids, saturated fats, sand dairy products and increasing the consumption of vegetables, fruits, legumes (beans, peas, and lentils), and whole grains reduces the risk of AD [11].

Dietary restriction (DR) has been demonstrated as a reliable and effective intervention to enhance the longevity and health status of various model organisms [12,13]. The principal DR regimens are reported in Table 1 and include caloric restriction (CR), intermittent fasting (IF) regimens, restriction of specific macronutrients, ketogenic diets (KD), and fasting-mimicking diet (FMD). All of them have been shown to reduce metabolic markers associated with aging in both animal and human studies [12,14,15].

In recent years, the FMD has gained attention and is considered to have significant potential among various DR regimes due to several reasons [16,17]. Unlike traditional fasting, FMD provides a controlled intake of essential nutrients, allowing individuals to experience some of the benefits of fasting while still obtaining vital nutrients. This feature reduces the concerns associated with long-term fasting, such as nutrient deficiencies. FMD offers a structured eating plan with specific meals and calorie restrictions, making it more feasible and potentially easier to adhere to compared to prolonged fasting protocols. FMD typically lasts for a few consecutive days—typically three to five—making it more manageable for individuals than longer fasting periods [16]. Due to all described properties, FMD could be a potential and feasible prevention and therapy option for AD. Thus, with this narrative review, we aimed to summarize the current evidence of the potential effects of FMD on AD as a preventive or curative strategy. A literature search in Pubmed, Medline, and Cochrane databases of all articles published with keywords “fasting”, “dietary restriction”, “fasting mimetics”, and “Alzheimer’s disease” was carried out. The keywords were used in all possible combinations to obtain the maximum number of articles. All studies, from bench and animal models to clinical, were included. After a short overview of the impact of dietary restriction and fasting mimetics diets on human health and longevity, we focused on studies that have examined the potential role of FMD in AD prevention or disease-modifying. The review concludes with a general discussion of the mechanisms underlying this unique trajectory and its implications for future research.

## 2. Nutrition, Metabolism, and Longevity: The Potential Role of Caloric Restriction

Genetic, environmental, and lifestyle factors strongly impact on human lifespan. Among them, nutrition is considered an essential component not only affecting health status but also with the potential to increase lifespan [18]. Nutrients are substances the body requires to perform its basic functions and are contained in the foods introduced with diet. They have three essential actions: provide energy, contribute to body structure, and regulate chemical processes [18,19,20]. These basic functions allow us to live, grow, reproduce, move, and respond to environmental insults. The main components of food are carbohydrates, fats, and proteins that are metabolized as a source of energy production (ATP, adenosine triphosphate) in cells [21]. Cells require chemical energy to drive metabolic reactions, and mitochondria represent the cell’s powerhouses due to their action in promoting the energy-releasing activities of electron transport and proton pumping with the strength-protecting method of oxidative phosphorylation to produce ATP from nutrients. Although biochemical energy production is required to sustain cell viability and functions, excessive energy production might also induce cells to damage. In fact, in this condition, the mitochondrial electron transport chain generates the superoxide anion radical (O2•^−^) through direct reactions with molecular oxygen. This superoxide anion radical then undergoes dismutation to form hydrogen peroxide (H_2_O_2_). Additionally, hydrogen peroxide has the potential to react further and produce the hydroxyl radical (HO) [21,22]. Reactive oxygen species (ROS) are generated by mitochondria both inside and outside the cell. In turn, these ROS have the potential to damage various components of mitochondria and initiate degradation processes. These toxic reactions play a significant role in the aging process and are a key principle of the free radical theory of aging. This theory suggests that the accumulation of oxidative damage caused by ROS contributes to aging. Much of the supporting evidence for this theory stems from the correlation observed between the levels of antioxidant defenses specific to different species and their lifetime energy expenditure [22].

Energy expenditure refers to the energy utilized by an individual to sustain vital body functions. The overall daily energy expenditure is influenced by factors such as the resting or basal metabolic rate (BMR), the thermogenic effect induced by food, and the energy expended during physical activity. Numerous studies have indicated that animals with longer lifespans exhibit lower BMR levels [23]. These data formed a cornerstone of the rate-of-living hypothesis and the free radical damage theory of aging, each of which recommends that the sturdiness of various animal species is inversely proportional to their power expenditure. Production of unfavorable reactive oxygen species from metabolism is a notion to purpose extra oxidative strain and decreased longevity. A notable correlation has been observed between individuals who are entirely functional and free of major medical conditions and their lower BMR in comparison to those with diseases and functional impairments. These findings reinforce the significance of health status in energy regulation, highlighting that an elevated BMR could potentially serve as a valuable biomarker for poor health among older individuals [24,25]. In this context, dietary composition and calorie level are key factors affecting BMR, aging, and age-related diseases [23].

CR, among DR regiments, is the only intervention that clearly reduces BMR and improves longevity [26]. CR generally refers to a 20–40% reduction in daily total calorie intake without lowering micronutrient intake, whereas dietary restriction refers to limiting a particular macronutrient (proteins, carbohydrates, or fats) with or without a reduction in total calorie intake. The health- and life-span-extending effects of CR have been characterized across distinct species, including yeast (Saccharomyces cerevisiae), nematodes (Caenorhabditis elegans), flies (Drosophila melanogaster), rodents (Mus musculus), and non-human primates (Macaca mulatta) [26,27,28]. CR has been proposed to enhance longevity by reducing the production of ROS and slowing down metabolism [29]. As a result, damage to redox-sensitive transcription factors is prevented, and activation of pro-inflammatory pathways is inhibited [29,30].

CR has been shown to positively affect multiple age-related changes in humans [31]. A study conducted on non-obese, healthy adults for twenty-four months using continuous CR found that it was safe and improved their quality of life. It also caused weight loss of 10–13%, primarily from fat mass, which stabilized after a year [32]. In addition, CR reduced fasting insulin levels, body temperature (a possible marker for metabolic rate), resting energy expenditure, oxidative stress, and thyroid axis activity [31,32]. In recent studies conducted in overweight humans, CR ameliorated many health outcomes, including reducing several cardiac risk factors [33,34,35], improving insulin sensitivity [36], and enhancing mitochondrial function [37]. Additionally, prolonged CR has been found to reduce oxidative damage to both DNA and RNA, as assessed in white blood cells [38]. Interestingly, humans undergoing CR experience a phenomenon known as “metabolic adaptation,” wherein the metabolic rate decreases below the expected level. This intriguing adaptation has been observed to potentially contribute to longevity in humans [39]. Moreover, in healthy subjects, CR can decrease the levels of circulating tumor necrosis factor-α as well as cardiometabolic risk factors (including high triglycerides, cholesterol, and blood pressure) [40,41]. Obese patients who undergo CR and achieve weight loss experience notable reductions in insulin growth factor-1 (IGF1) levels and improved insulin resistance [42]. In this context, the population of Okinawa Island in Japan has garnered attention due to their well-documented good health and a remarkable number of centenarians. These characteristics have been attributed, in part, to their dietary pattern, including a mild and consistent CR [43]. 

From a biological point of view, CR downregulates the expression of many genes involved in oxidative stress and reduces oxidative damage in numerous tissues [31]. Other biological changes associated with CR contributing to the observed increases in health span and longevity include autophagy (“self-eating” of damaged organelles) and the maintenance of functional mitochondria through biogenesis (generation of new mitochondria) [44]. Despite these health-promoting biological changes, most individuals have difficulty engaging in CR for a long time. Moreover, adherence to long-term CR regimens is often hindered by psychosocial difficulties, which are, at least in part, offered by modern society through high-caloric diets. Thus, the idea of CR mimicry has been advanced as an opportunity for fasting or CR regimens. A periodic, short-term, low-calorie, and low-protein dietary intervention—in line with the health advantages of low-calorie or low-protein diets—has been developed to overcome the fact that many fasting-based interventions are likely not feasible or extremely difficult to adhere to.

The FMD is one nutritional opportunity technique that could produce comparable organic modifications as a CR that has received increasing interest from the scientific community. FMD permits decreased calorie consumption in preference to whole meals abstinence throughout the fasting period. FMD constitutes periodic cycles of consecutive days consuming a reduced-calorie diet followed by eating ad libitum [16,17]. FMD has emerged as a dietary modification that could benefit cardiometabolic diseases and weight loss programs [16,17,45].

## 3. Fasting-Mimicking Diet: What Evidence for Health Status

FMD is a highly studied cyclic variant of CR designed to induce metabolic responses related to fasting through a low-calorie diet. FMD is a plant-based diet, low in protein and sugar but relatively high in fat. The FMD has been developed to be used in periodic cycles from every two weeks to every several months, lasting from four to seven days for humans and two to five days for mice. The five-day human FMD affords about 55% of the endorsed day-by-day calorie consumption on day one and 35% on the following days, two–five. The short duration (five consecutive days per month recommended for humans) and the periodic application are thought to improve adherence and reduce dietary “fatigue,” thus enabling easy inclusion into existing lifestyles. There are no subtypes or variations within this specific protocol.

From studies conducted in mice, it has been shown that FMDs extend the median lifespan, reduce inflammation and most cancer incidence, keep cognitive performance, and improve overall health [46,47,48,49,50]. Choi and colleagues [50] demonstrated that periodic three-day cycles of FMD effectively improve demyelination and related symptoms in a mouse experimental autoimmune encephalomyelitis model. In detail, the FMD reduced clinical severity in treated mice and reversed most symptoms in 20% of animals. Most importantly, these improvements were associated with higher corticosterone and regulatory T (Treg) cell numbers and lower levels of pro-inflammatory molecules. One interesting effect of cyclic FMD was the induction of atrophy/quiescence followed by vigorous regeneration and stem cell activation in several tissues [50]. Similarly, metabolic parameters, which include blood glucose, insulin, and IGF-1, were reduced during the FMD phase and returned to control levels during a regular feeding regimen. A further study tested an FMD in db/db mice (a genetic model of type 2 diabetes) [47]. Such a diet was given every other week for a total of eight weeks. The FMD was able to normalize blood glucose levels, with significant improvements in insulin sensitivity and β cell function. In fact, the deterioration of pancreatic islets and the loss of β cells in diabetic mice were prevented. Of interest, the FMD was also associated with a reduction in hepatic steatosis [47].

However, evidence of FMD for human health is still poor. A study was conducted to test the effects of FMD on markers and risk factors associated with aging and age-related diseases [46] and involved one hundred healthy participants who were divided into two study groups. Subjects who followed three months of an unrestricted diet and subjects who consumed the FMD for five consecutive days per month for three months. Three FMD cycles decreased body weight and total body fat, reduced blood pressure, and lowered IGF-1 serum levels. No adverse effects were reported. After three months, subjects under the control diet were crossed over to the FMD, resulting in a total of seventy-one subjects completing three FMD cycles. A post hoc analysis of subjects from both FMD arms showed that fasting glucose, IGF-1, triglycerides, total cholesterol, C-reactive protein, body mass index, and blood pressure were more significantly lowered in participants at higher risk of cardiometabolic disease. With this study, the authors also showed that cycles of a five-day FMD are safe, feasible, and effective in potentially reducing risk factors for age-related diseases [46].

## 4. Alzheimer’s Disease and Dementia: The Metabolic Pathway

In AD, there is growing evidence supporting the involvement of metabolic pathways in the development and progression of the disease. The metabolic hypothesis of AD suggests that disruptions in various metabolic processes within the brain contribute to the pathology of the disease. One key metabolic pathway that has been extensively studied in relation to AD is glucose metabolism [51,52]. The human brain utilizes around 20% of the body’s energy resources, representing about 2% of the body’s mass [52]. This underscores a huge metabolic workload, which is mainly fuelled by glucose. Strong evidence shows that inadequate supply and utilization of energy sources to the brain can lead to impaired energy production and metabolic dysfunction [9]. Insulin is a hormone that plays a crucial role in regulating glucose metabolism and promoting cellular glucose uptake. While insulin is primarily known for its role in regulating glucose metabolism in the body, it is also produced and utilized in the brain. Many studies have suggested that insulin levels may be elevated in the brains of individuals with AD, indicating a state of compensatory response or resistance to insulin. This phenomenon is referred to as “brain insulin resistance”. Brain insulin resistance can lead to impaired glucose uptake and utilization in brain cells, resulting in reduced energy production. This energy deficit can contribute to the dysfunction and degeneration of neurons. Moreover, insulin signaling pathways play a role in various cellular processes, including neuronal survival, synaptic plasticity, and clearance of toxic proteins. Disruption of insulin signaling in the brain may also contribute to the accumulation of pathological proteins, such as Aβ, and the development of AD. Accordingly, amyloid plaques may first appear in brain regions that are characterized by high levels of aerobic glycolysis in patients with AD [53]. White matter loss and oligodendrocyte dysfunction represent early brain changes in AD [54,55,56], suggesting that the supply of energy substrates is reduced, resulting in neuronal dysfunction. Thus, Aβ accumulation and tau hyperphosphorylation may lead to further disruption of myelin integrity and oligodendrocyte maturation and metabolism through oxidative stress and neuroinflammation [57,58,59,60]. Like oligodendrocytes, astrocytic morphology and functions are abnormal in AD, resulting in deleterious effects, including reduced carbon delivery to neurons for oxidative phosphorylation and dysregulated linkages between neuronal energy demand and regional blood supply [61]. Similarly, microglial metabolism shifts from oxidative phosphorylation to aerobic glycolysis as observed upon acute exposure to stressors such as fatty acids or Aβ [62]. Taken together, these data suggest that changes to cellular bioenergetics may be considered key factors that influence the onset of AD pathophysiology [52,63]. However, many questions remain about the causal role of individual cell type bioenergetics and AD, and future studies should aim to define these metabolic changes more clearly over the entire AD time course.

Metabolic pathways and diet are intricately linked, as the nutrients we consume through our diet serve as the building fuel for various metabolic processes in our body. Moreover, different dietary components can also influence the regulation of metabolic pathways. Dietary factors like high sugar or fat intake can affect insulin sensitivity and disrupt metabolic homeostasis.

Conversely, specific diets, such as low carbohydrates, can alter metabolic pathways by promoting the utilization of fat as the primary energy source. This shift in metabolism can have implications as potential therapeutic applications in certain medical conditions. The influence of diet on the brain is a complex process that is influenced by various factors. These factors can impact the transportation and metabolism of nutrients, affecting their availability to the brain. Recent findings from neuroimaging studies in humans suggest that different dietary patterns may have an impact on cerebral bioenergetics, potentially starting in middle adulthood, even before any observable effects on other biomarkers related to AD and changes in brain structure become apparent [64]. In the past three decades, growing evidence has indicated the advantageous outcomes of fasting and CR as viable approaches, either independently or in combination with other lifestyle interventions like physical activity, and as potential complements to pharmacological treatments in the prevention and treatment of AD [65].

## 5. Alzheimer’s Disease: From Caloric Restriction to Fasting-Mimicking Diets

A long literature story shows that different dietary patterns may be effective in limiting AD progression in mouse models and fasting [65]. Significant alterations in neurochemistry and neuronal network activity occur because of fasting, particularly in crucial brain regions like the hippocampus, striatum, hypothalamus, and brainstem. At the molecular level, various signaling pathways have been discovered, facilitating structural changes such as enhanced synaptic density and neurogenesis, as well as functional adaptations in neuronal circuits in response to nutrient restriction, specifically low glucose levels. Previous research performed in PS1 mutant knockin mice (a model of familial AD) concluded that an alternate-day fasting (IF) routine of three months was able to reduce excitotoxic damage to hippocampal CA1 and CA3 neurons when compared with mice fed at ad libitum. Again, CR for fourteen weeks in amyloid precursor protein (APP) and PS1 transgenic mice was associated with a reduction in the accumulation of Aβ plaques and decreased Aβ plaque-associated astrocyte activation [66]. Furthermore, CR diets in other AD mouse models slowed the progression of Aβ deposition in the hippocampus and cerebral cortex [67]. A study conducted over a period of seven to fourteen months on 3xTg mice—which are often used to investigate synaptic dysfunction and AD-related pathology—found that CR improved age-related behavioral deficits [68]. Again, after four months of protein restriction cycles, alternated with regular feeding, 3xTg male mice exhibited improved behavior performance and reduced phosphorylated tau compared with ad libitum-fed animals, associated with a reduction in IGF-1 signaling during the restricted period [69]. Again, CR exhibits a remarkable ability to mitigate oxidative stress, a key factor implicated in cognitive decline. By minimizing the harmful effects of oxidative stress, CR fosters a conducive environment for optimal cognitive performance, thereby delaying aging-related cognitive decline [70]. Furthermore, CR has been shown to stimulate the proliferation of new neurons, a phenomenon known as neurogenesis in various brain regions, including the hippocampus. Neurogenesis plays a crucial role in maintaining cognitive function by facilitating learning, memory, and overall brain plasticity. Thus, the pro-neurogenic effects of CR contribute to its ability to preserve cognitive function during aging (reviewed in Moharajan [71]). More recently, novel evidence has emerged from a study conducted in a rat model of Alzheimer’s disease (AD) induced by aluminum chloride (AlCl_3_). The results of the study demonstrated that the induction of AD with AlCl_3_ led to cognitive and behavioral deficits, impaired autophagy, increased apoptosis, and disrupted neurogenesis and astrocyte activation in the hippocampus. Caloric restriction (CR) showed neuroprotective effects against these changes induced by AlCl_3_, partially ameliorating the behavioral, cognitive, biochemical, and histological alterations. CR exhibited significant improvements in behavior and histology [72].

Studies on long-term fasting in humans have been limited. A recent systematic review and preliminary meta-analysis, including eleven trials on the efficacy of dietary restriction on cognitive function, showed that DR has varying degrees of positive effect on cognitive function in overweight/normal-weight people and subjects affected by Mild Cognitive Impairment [73]. Some studies have specifically investigated the association between CR and AD in humans, primarily through observational and epidemiological research [74]. Consuming a diet with lower calories significantly reduces the risk of AD [75]. Another study followed a group of participants over the age of 60 for an average of 4.5 years and found that CR was associated with a reduced risk of cognitive decline [76]. These studies have shown some promising results, indicating that CR may be associated with a lower risk of developing AD and improved cognitive function.

Witte et al. demonstrated an improvement in verbal memory scores after 3 months of caloric restriction in healthy study participants with an average age of 60 years [77]. Leclerc and collaborators replicated these cognitive benefits in a larger sample, multicenter randomized controlled trial involving 220 healthy non-obese adults [78]. The study revealed that a daily 25% reduction in caloric intake over a period of two years resulted in a substantial improvement in working memory [78]. However, the specific effects of fasting remained unknown. In a longitudinal study focused on healthy aging, older adults with mild cognitive impairment were followed up [79]. The study included 99 participants aged over 60 years, of whom 37 practiced intermittent fasting for two days per week from dawn to sunset, 35 observed fasting for 12 months, and 27 did not practice fasting. The group that regularly practiced fasting showed a significantly higher proportion of participants with successful aging, defined as being free of common chronic diseases, having a Mini-Mental State Examination score above 22, good functional ability, and a good quality of life (24.3% in the fasting group vs. 3.1% in the non-fasting group) [79].

There is limited evidence on the effects of FMD on AD. Only one study is available so far [80]. In this study, the authors demonstrated that FMD cycles have a more significant effect on reducing AD pathology and cognitive decline than cycles of protein restriction in mouse models of AD (E4FAD and 3xTg). The E4FAD model carries both the human apolipoprotein E4 (APOE4) gene variant and gene mutations associated with familial AD. The triple transgenic 3xTg model carries three genetic mutations that are commonly found in familial AD: human APP (amyloid precursor protein) mutation, human PS1 (presenilin 1) mutation, and human tau mutation. Four months of bi-monthly FMD cycles in female E4FAD mice mitigate Aβ hippocampal and cortical load, reduce Aβ38/40/42, and increase IL-2 (interleukin-2) expression in cortex extracts. The behavioral tests suggest that bi-monthly FMD cycles improve visual attention, working memory, and spatial memory; ameliorate anxiety-associated behaviors; and increase exploratory activity in E4FAD female mice. In the 3xTg mice, long-term FMD cycles reduce amyloid-beta (Aβ) accumulation and hyperphosphorylated tau in the hippocampus. Additionally, FMD cycles promote the generation of new neural stem cells, decrease the number of microglia (immune cells in the brain), and lower the expression of genes related to neuroinflammation, including NADPH oxidase (Nox2), which produces superoxide, enhancing oxidative stress. Combined findings from these two mouse models of AD suggest that cycles of FMD can decrease the levels of pathological indicators, such as Aβ and hyperphosphorylated tau. FMD also appears to reduce microglia density and markers associated with neuroinflammation, ultimately leading to improved cognition. The outcomes observed in the 3xTg/Nox2-KO mice and with the use of apocynin treatment further support the hypothesis that FMD cycles exert positive effects on the 3xTg and E4FAD models by influencing the activity of microglia and potentially brain macrophages. This modulation enables them to undertake protective functions, including the clearance of Aβ, while simultaneously reducing oxidative stress levels [80].

To assess the feasibility and safety of fasting-mimicking diet (FMD) cycles in patients diagnosed with amnestic mild cognitive impairment (aMCI) or mild AD, the same group is performing a phase I/II clinical study. The study involved patients with aMCI or mild AD who had good nutritional status. The trial was randomized and placebo-controlled, with single-blinding. After screening and a baseline assessment of cognitive performance, functional status, and caregiver burden, subjects were randomly assigned to the FMD (active) or placebo group. The placebo group received a diet in which one meal (lunch or dinner) was replaced with a pasta or rice-based meal with vegetables for five days a month, without any supplements. On the other hand, the FMD group completed five-day FMD cycles, accompanied by supplements known for their fasting-mimicking, neuroprotective, anti-inflammatory, and antioxidant properties. These supplements included olive oil, coconut oil, algal oil, nuts, caffeine, and cocoa. Additionally, the FMD group received these supplements between FMD cycles for twenty-five days while following a regular diet. The effectiveness of interventions relies heavily on ensuring adherence to dietary changes. In general, adherence to the prescribed diet has been satisfactory, even during the periods between FMD cycles when patients consume multiple supplements throughout the day (detailed data are still unavailable). These preliminary findings suggest that administering five-day FMD cycles monthly has been feasible and generally safe. No data are still available on cognitive performances [80]. As far as feasibility is concerned, a recent study investigating the effect of FMD on metabolic health factors in patients with prostate cancer showed that most of the subjects reported the program was easy to undertake, resulting in an 83% compliance to three cycles, a 91% compliance to two cycles, and a 100% compliance to one cycle. Crucially, no adverse effects associated with FMD were observed [81].

## 6. Summary and Final Remarks

Figure 1 summarizes the main mechanisms implicated in AD pathology and modulated by two DR regimens, CR and FMD. The present review suggests that FMD may have potential benefits for AD. Studies conducted in animal models of AD have shown promising results regarding the effects of FMD cycles on reducing cognitive decline and AD pathology. Moreover, FMD was associated with lower amyloid plaque accumulation, improved metabolic health, modulation of inflammation, and enhancement of neural stem cell production. Clinical trials evaluating the feasibility and safety of FMD cycles in individuals with aMCI or mild AD have also been initiated.

Preliminary findings suggest that FMD cycles administered once a month have been feasible and generally safe. Adherence to the FMD regimen and consumption of the prescribed supplements have been satisfactory. In terms of adherence, the FMD may offer certain advantages compared to other caloric restriction programs. The FMD is followed for a relatively brief period, which may make it more feasible and sustainable for individuals compared to longer-term caloric restriction programs. Unlike complete fasting, the FMD allows for some calorie intake, which may help individuals adhere to the diet more effectively compared to programs that involve complete deprivation of food. The FMD provides a specific meal plan that outlines what and when to eat, which can help individuals with adherence by providing clear guidelines and structure. The FMD has been associated with various health benefits, such as improved metabolic markers, reduced inflammation, and potential longevity benefits. These potential benefits may provide additional motivation for individuals to adhere to the diet.

While such a diet offers numerous potential health advantages, it is important to be aware of its associated transient side effects, including dizziness, headache, fatigue, and general weakness. Thus, further monitoring and recruitment of subjects are necessary to confirm the safety and efficacy of FMD, especially in older persons. However, as a preventive and curative strategy, the FMD may have the maximum effectiveness during the preclinical phase of the disease. In this disease stage, with the presence of a caregiver, it is possible to hypothesize a higher adherence compared to the dementia stage when the brain damage is already advanced. Above all, it can be practically hard to implement this diet in an individual with dementia, especially with behavioral impairments. In conclusion, while the FMD holds promise for improving cognitive function in AD, more research is needed to fully understand its effectiveness and safety.

### Key Points

The Fasting Mimicking Diet (FMD) is a program that aims to mimic the effects of fasting while still allowing some food intake.The FMD involves consuming a low-calorie, low-protein, and low-carbohydrate diet for 4–7 days.Alzheimer’s disease (AD) is a progressive neurodegenerative disorder characterized by the accumulation of amyloid plaques and neurofibrillary tangles in the brain, leading to cognitive decline.Animal studies have suggested that fasting can reduce the levels of amyloid beta in the brain, a key component of amyloid plaques.Limited human studies have found that the FMD may improve cognitive function in patients with mild cognitive impairment, a precursor to AD.While the FMD holds promise for improving cognitive function in AD, its effectiveness and safety require further investigation.

## Figures and Tables

**Figure 1 biomolecules-13-01133-f001:**
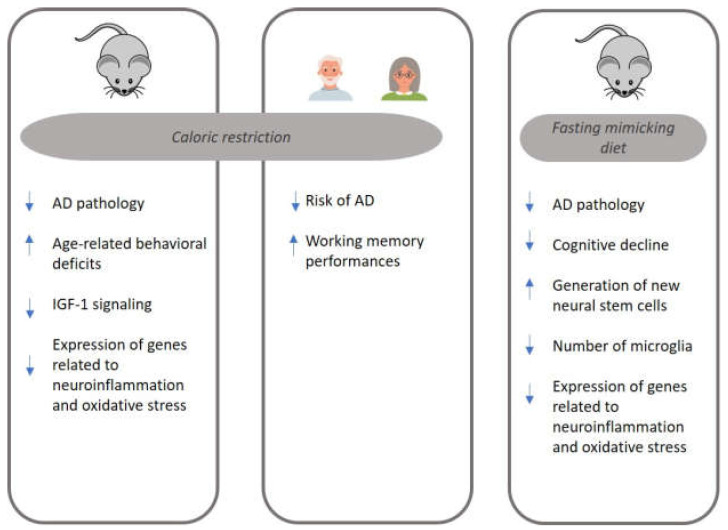
The main mechanisms implicated in Alzheimer’s disease (AD) pathology and modulated by caloric restriction (CR) and fasting-mimicking diet (FMD). AD: Alzheimer’s disease; IGF-1: Insulin Growth Factor-1.

**Table 1 biomolecules-13-01133-t001:** Principal dietary restriction (DR) regimens.

*Caloric restriction (CR)*	Throughout the entire duration of the dietary intervention, participants have successfully implemented a reduction in caloric intake by 20–30% below the average, ensuring they maintain adequate nutrition and avoid any risk of malnutrition.
*Intermittent fasting (IF)*	Alternating periods of fasting and eating. IF includes: Time-restricted feeding (TRF): This IF regimen involves limiting the daily eating window to a specific time, typically 8–10 h, and fasting for the remaining 14–16 h. Alternate day fasting (ADF): ADF involves alternating between fasting days and regular eating days. On fasting days, calorie intake is severely restricted or completely avoided, while on eating days, individuals can consume their usual amount of food. Modified fasting: This approach involves reducing calorie intake on specific days of the week, often referred to as “fasting days,” while following a normal eating pattern on the remaining days. Typically, individuals consume a limited number of calories (e.g., 500–600 calories) on fasting days. 5:2 diet: This IF regimen involves eating normally for five days of the week and restricting calorie intake to around 500–600 calories on two non-consecutive fasting days. On the remaining days, individuals follow their regular eating patterns. Periodic fasting: Periodic fasting involves longer fasting periods ranging from 24 h to several days. For example, individuals may fast for 24 h once or twice a week or opt for longer fasting periods of 48 or 72 h intermittently.
*Restriction of specific macronutrients*	Glucose and carbohydrate restriction; Protein restriction; amino acid restriction; Micronutrient restriction.
*Ketogenic diets (KD)*	A high-fat, low-carbohydrate dietary approach that aims to induce a state of ketosis in the body. The typical macronutrient distribution involves consuming a very low amount of carbohydrates (generally less than 50 g per day or 5–10% of total calories), a moderate amount of protein, and a high proportion of dietary fat (70–75% of total calories).
*Fasting mimicking diet (FMD)*	A dietary protocol designed to mimic the effects of a prolonged fast providing some nutrient intake. The main components of an FMD typically include consuming plant-based foods such as vegetables, nuts, and seeds and healthy fats like olive oil. The macronutrient distribution is calculated to provide around 40–50% of normal calorie intake on the first day and around 10–20% for the following days of the fasting period (4–7 days every 15–365 days)

## Data Availability

Data sharing not applicable.

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
