# Peer review of "The Potential of Fasting-Mimicking Diet as a Preventive and Curative Strategy for Alzheimer’s Disease"

_biomolecules, 2023, doi:10.3390/biom13071133_

Round 1

Reviewer 1 Report

The review addresses the potential of FMD in preventing and treating AD. The review first describes the different dietary restriction strategies and then it summarizes the studies that relate CR with age-related changes (such as ROS production, energy expenditure, insulin sensitivity, mitochondrial function...).  Subsequently, the review is focused specifically on the impact of FMD in health, using mainly mice studies, due to the lack of great evidence for FMD in healthy humans. Yet, it is included the only study that explores FMD in healthy subjects. Finally, it is focused on AD and describes the effect of dietary restriction (CR and FMD) in AD mouse models and clinical trials in humans (MCI, AD).

The topic is original and comprises recent literature that supports the potential role of FMD in improving AD pathology (DOI: 10.1016/j.celrep.2022.111417; DOI: 10.1039/c9fo02611h).

Although other reviews have been published addressing the molecular mechanisms of dietary restriction that promote health and longevity (DOI: 10.1016/j.arr.2016.10.005; https://doi.org/10.1038/s41580-021-00411-4; DOI: 10.1146/annurev-nutr-122319-034601; DOI: 10.3390/nu14235086), this review summarizes the studies that explore the association between CR/FMD and AD in mouse models and humans.

The review concludes that FMD has shown promising results in terms of reduction of cognitive decline and AD pathology in animal models of AD and, therefore, it may have potential benefits for AD studies. It also mentions the clinical trial that has been initiated to evaluate the safety of FMD in individuals with MCI and AD.

Therefore, the conclusions are consistent with the results presented in the review.

The review includes the most recent literature, and the references are appropriate.

Table 1 which contains a summary of the different DR regimens is well structured.

There is exclusively one final figure that comprises a summary of the mechanisms involved in AD and their modulation in response to FMD and CR. The figure is clear and well organized.

As a minor point, some “typo” mistakes should be corrected along the text.

Good quality of English.

Author Response

We express our sincere gratitude to the Reviewer for valuable suggestions and feedback, which have greatly contributed to the revision and improvement of our manuscript. 

Thank you for your feedback. We have carefully reviewed the manuscript and made the necessary revisions.

Reviewer 2 Report

The manuscript by Boccardi et al entitled "The Potential of Fasting-Mimicking Diet as a Preventive and Curative Strategy for Alzheimer's Disease" is relatively well written. The manuscript studies dietary methods to improve cognition with aging. In general, the manuscript is well-written, except section 2 on page 3 which is somewhat incomprehensible. In the discussion some more space should be dedicated to the different forms of FMD that exist, and the putative differences in outcome of these FMD strategies. Similarly, can this approach be used in what AD patients, MCI or earlier or later, especially the latter seems hard to do. Some comments about adherence to the diet might also be helpful. Especially possible negative effects of the diet should be discussed.

section 2 on page 3 which is somewhat incomprehensible.

Author Response

We express our sincere gratitude to the Reviewer for  valuable suggestions and feedback, which have greatly contributed to the revision and improvement of our manuscript. 

Thank you so much. Please find below the point by point response.

  1. Section 2 on page 3 has been appropriately revised based on your suggestion.
  1. Dear reviewer, the fasting mimicking diet (FMD) is a dietary restriction regimen (see Table 1) designed to mimic the effects of fasting while still allowing for some calorie intake. It involves following a specific meal plan for a set number of days, typically ranging from five to seven consecutive days every 15–365 days, followed by long periods on a normal diet. The FMD is characterized by a low-calorie and low-protein intake, while providing adequate amounts of healthy fats, complex carbohydrates, and micronutrients. The macronutrient composition of the diet is carefully designed to induce metabolic and cellular changes similar to those observed during periods of fasting. In mice, four-day bi-monthly cycles of the FMD started at middle age extend longevity, reduce tumors by nearly 50%, decrease inflammatory diseases, and increase cognitive performance in old age. In humans, three monthly cycles of a five-day FMD reduce markers or risk factors for aging, diabetes, cancer, and cardiovascular disease, including cholesterol, blood pressure, CRP, IGF-1, and fasting glucose, particularly in subjects with elevated levels of these markers at baseline. The FMD specifically refers to the fasting mimicking diet developed by Dr. Valter Longo, a renowned researcher in the field of aging and nutrition. Dr. Longo has conducted extensive research on the effects of fasting and developed a specific protocol for the FMD known as ProLon. There are no subtypes or variations within that specific protocol (please see doi: 10.1093/geroni/igac059.362). We have further clarified this concept in the revised text.
  1. As kindly suggested, comments about adherence as well as possible negative effects have been added and discussed.

Reviewer 3 Report

The manuscript presented by Boccardi et al. reviews the state of the art of utility of FMD in Alzheimer's disease. However, most of the manuscript does not discuss AD, and does not justify why this diet is chosen for the review and not others. Moreover, only one publication is reviewed in the review of that type of diet [74].
Minor issues

Line 61: are probably risk factor for AD not for aging
Table 1 the second kind of diet is not explained
Line 80: glucose metabolism is different from FMD

Section 2 is not about AD, is about longetivity. These two terms are related but this relationship is not expleaines.

Section 3. FMD should be focus on AD

Sections 4 and 5 (AD) are poorly explained

Section 5 only one paper is reviewed

Author Response

We express our sincere gratitude to the Reviewer for  valuable suggestions and feedback, which have greatly contributed to the revision and improvement of our manuscript.

Please find beloww a point-by-point response. Thanks again.

Response 1: Thank you so much for your comments and observations. The review, as structured, addresses the potential of FMD in preventing and treating AD. The review first describes the different dietary restriction strategies and subsequently focuses specifically on the impact of FMD on health, primarily using mice studies due to the lack of extensive evidence for FMD in healthy humans. Thus, we have included the only study that explores FMD in healthy subjects. Finally, the focus is on AD, describing the effect of FMD on AD mouse models and clinical trials in humans (MCI, AD).

Many other reviews have been published addressing the molecular mechanisms of various dietary restriction regimens that promote health and longevity. Therefore, we aimed to explore the association between FMD, health, and AD. The main characteristics of the fasting mimicking diet (FMD) include caloric restriction, time-limited duration, macronutrient composition, nutrient density. Regarding adherence, the FMD may offer certain advantages compared to other caloric restriction programs: short duration, controlled caloric intake, structured meal plan, potential health benefits.

All these aspects have been properly added in the revised manuscript.

Response 2: “All of them have been shown to reduce metabolic markers/risk factors associated with aging in both animal and human studies”. With this sentence we just wanted to refer to metabolic markers of aging. Thus we removed “risk factors” to don’t make confusion to the reader.

Response 3:  The second type of diet (with all variants) is reported in the Table 1. IF refers to alternating periods of fasting and eating. IF includes: Time-restricted feeding, Alternate day fasting, Modified fasting, 5:2 diet, Periodic fasting. All of them are reported and described in the table.

Response 4: Sorry in line 80 a mistake occurred. We properly revised the sentence.

Response 5: Thank you very much for this observation. Yes, in section 2, as mentioned in the introduction, we first describe the different dietary restriction strategies, starting from metabolism and longevity, to further explain the link between FMD (a type of dietary restriction regimen) and human health, as well as its involvement in AD.

Response 6: In the section 3 we explored evidence of FMD for health status leading the reader to the metabolic hypothesis of Alzheimer's disease further described in the sections 4 and 5.

Response 7: Section 4 and 5 have been properly better explained and improved.

Response 8: In the section 5  it is included the only study, so far,  that specifically explores FMD in subjects affected by AD and in particular in MCI and mild dementia.

Thanks again for your efforts. 

Round 2

Reviewer 2 Report

The manuscript has revised appropriately.

Author Response

Dear Editor,

Thank you so much.

Regards

Reviewer 3 Report

In my opinion, the problems are the same as in the previous version. The manuscript is not suitable for publication, as it reviews only one paper (FMD-AD relationship). There are not enough papers on the topic to review. Most of the manuscript deals with general issues, not related to AD, and neither related in the paper. There are no conclusions of interest beyond those of the paper reviewed.

Author Response

Thank you so much for the kind response. We would like to express our sincere gratitude to the Editor and Reviewers for their valuable suggestions and feedback, which have greatly contributed to the revision and improvement of our manuscript. Their input has been instrumental in refining our study and enhancing its overall scientific merit. 

As stated by the Editor, the article describes how dietary modifications can reduce our chances of developing of AD and reduce progression of AD. The article included brief description of various types of dietary restricted regimens  and laid special emphasis on Fasting mimicking Diet (FMD) and provided an explanation to do so. Editor further stated that the flow and sequence of article is very clear and conveys the idea. I am sorry that you consider our article not suitable for publication. This is a relatively newer area of intervention in AD using FMD. However, the review has included all relevant pre-clinical mice studies and what available in humans. Please find attached our revised version upon other suggestions. Thanks again for your time and efforts.